# Fog Monitoring through Machine Learning of Signal Attenuation Data from Microwave Links from Cellular Communication Networks

Indy van Grinsven<sup>1</sup>, Meiert Willem Grootes<sup>2</sup>, Remko Uijlenhoet<sup>3</sup>, Gert-Jan Steeneveld<sup>1</sup>.

<sup>1</sup>Wageningen University, Meteorology and Air Quality Section, PO box 47, 6700 AA Wageningen, The Netherlands.

<sup>2</sup>NLeScienceCenter, Matrix THREE, Science Park 402, 1098 XH Amsterdam, The Netherlands.

<sup>3</sup>Delft University of Technology, Faculty of Civil Engineering and Geosciences, Department of Water Management, Delft, The Netherlands.

Correspondence to: Gert-Jan Steeneveld (Gert-Jan.Steeneveld@wur.nl)

Abstract. Fog poses significant challenges in various sectors, from transportation safety to water resource management. Traditional fog detection methods rely on limited monitoring capabilities, hampering forecasting and nowcasting. This study investigates the potential of machine learning in fog classification based on microwave link signal attenuation data, utilizing existing commercial cellular communication networks. Using data from The Netherlands, the study explores machine learning models using the McFly model architecture. By incorporating multiple predictors including Received Signal Level (RSL) data, trends, and time variables, the models aim to distinguish fog from other weather phenomena. The research extends to a broader dataset from a commercial cellular network by using a reduced model and evaluates the feasibility of applying the reduced model on a larger scale. Results indicate promising prospects for machine learning in fog detection, with the Inception Time architecture showing notable accuracy in fog classification. However, challenges remain in balancing long and short-term data to align with fog evolution and reliably to distinguish fog from precipitation. Furthermore, the study suggests exploring higher-frequency telecommunication links for enhanced fog detection systems, emphasizing the need for continuous advancements in this domain.

#### 1 Introduction.


Fog occurs when the air at or near ground level reaches its dew point temperature and water can condense into a near-surface cloud, specifically reducing visibility below 1 km (AMS Glossary, 2020). Reduced visibility can lead to an increased risk of accidents and collisions on roads and highways, disrupt and cancel air traffic (Fabbian et al., 2007; Leander, 2010; Bari et al, 2024). The human and economic cost can even be comparable to that of other extreme weather events (Gultepe et al., 2007). In regions facing water scarcity, however, fog harvesting can provide fresh water for afforestation and drinking water (Jacobs et al., 2004), making it particularly valuable (e.g. Bintein et al., 2023).

Current numerical weather prediction models continue to face challenges in accurately predicting the timing, thickness, and dissipation of fog (Steeneveld, et al., 2015; Steeneveld and De Bode, 2018; Boutle et al., 2022; Antoine et al., 2023) and therefore weather forecasts rely heavily on nowcasting (e.g. Ribaud et al, 2021). However, the limited monitoring capability



and local nature of routine weather stations is a concern. Expanding this network to the appropriate density would be prohibitively expensive (Ellrod, 1995; Gultepe et al., 2007; van der Meulen, 2007; David et al., 2015 (henceforth D15)).

To facilitate the nowcasting, weather services currently use both in situ observations (temperature, humidity, downwelling radiation) and remote sensing information, and methods such as rule-based forecasting. Although both of these proxies can be used to detect and forecast the occurrence of fog, they are not perfect. The relation between fog and dewpoint temperature is not universal as other variables, notably wind speed, air quality, and turbulence, affect fog formation (Roach et al., 1976; van der Meulen, 2007).

Remote sensing offers alternative data through ground-based LIDAR-Ceilometers and satellite-based measurements. LIDAR-Ceilometers offer extensive fog data for the locations where they are installed, but the network density is relatively low (van der Meulen, 2007). Satellite-based remote sensing does offer widespread coverage and the ability to track the spatial development of fog. However, it is unable to distinguish low-level clouds from fog, and its view can be blocked by higher-level cloud layers (van der Meulen, 2007).

Ideally, one would use a widespread network of in situ fog detectors to reliably track fog occurrence and development. For precipitation, particularly rain, this has been achieved through the opportunistic use of existing measurements from wireless communication systems, specifically by tracking attenuation of the microwave signals used to transfer data between commercial cellular communication towers (Messer et al., 2006, Leijnse et al., 2007, Chwala et al., 2012; Overeem et al., 2013; Rios Gaona et al., 2018). Only limited research has been done to use this existing infrastructure as a fog detection system (David et al. 2013b (henceforth D13b), D15). However, even if fibre optics is currently replacing part of the wireless backhaul network, the Netherlands alone still contains thousands of cell towers equipped with these microwave links. Their distribution, near-ground-level (tens of meters) deployment, and pre-existence illustrates their potential as a dense country-wide network of visibility sensors. Also world-wide there is an estimated 5 million commercial microwave links available (Overeem et al., 2023).

D13b showed that fog droplets cause attenuation of the microwave signals used to transfer data between commercial cellular communication towers. Monitoring the disturbances in these microwave links has been suggested as a possible tool for real-time fog prediction. Unfortunately non-fog precipitation such as rain has a similar or higher attenuation effect on microwave links, which hampers distinguishing fog from other types of precipitation (David, et al., 2013a (henceforth D13a), D15). Therefore D13b excluded precipitation events in their dataset. Alternatively, D15 proposed to use microwave links at frequencies around 80 GHz (E-band) instead of 20–40 GHz ( $K_u/K_a$ -band). They hypothesized that at this frequency it would be possible to distinguish between fog and other precipitation types due to differences in the attenuation spectrum. Although E-band microwave links are becoming more common, the lower frequency microwave links are still more widespread. The real challenge then is to distinguish fog not only from clear weather but also from non-fog precipitation.

This study explores the use of machine learning (ML) as a tool for fog classification based on microwave link signal attenuation data. With enough real-time microwave link data available, for any given link the problem consists of a space-time classification. ML has proven itself as an effective tool for this type of analysis in the past (Fawaz et al., 2019; Zhao et al.,

https://doi.org/10.5194/egusphere-2025-2634 Preprint. Discussion started: 11 November 2025

© Author(s) 2025. CC BY 4.0 License.

2017). The hypothesis is that ML will be able to distinguish fog from precipitation by the difference in their respective

attenuation strength and variability. The study focuses first on analysis and ML model development for ~1.5 years of

attenuation data from one microwave link in the Dutch town of Wageningen. A number of simple ML architectures are

explored, as well as more deep learning architectures created using the Netherlands eScience Center's McFly python package

(Van Kuppevelt et al., 2020), which supports automated deep learning/ML based time series classification. The findings for

the link in Wageningen were then applied to a larger dataset of almost 3000 microwave links from the commercial T-Mobile

cellular network across The Netherlands between 30-05-2012 and 01-09-2012.

First, we report on the creation of a fog reference dataset (Section 2). Second, we explore the viability of using ML as a tool

for fog classification based on all available RSL data for one link over Wageningen through the creation of a complex neural

network (Section 3.2). Third, this classification is repeated though for a reduced and more realistically available amount of

data, and a reduced neural network, to test what is possible with currently available data (Section 3.3). Finally, in the interest

of empirical exploration, this study applies the reduced neural network to a large group of existing commercial microwave

communication links (Section 3.4).

2 Material and methods

2.1 Study area

80

The study area consists of the path between two university buildings in the town of Wageningen (The Netherlands), which

stretches 2.2 km, mainly over terraced houses (LCZ6; Stewart and Oke, 2012), a fairly typical Dutch suburban area (See Figure

1). Van Leth et al. (2018a, henceforth VL18) describe detailed primary and secondary data available from several microwave

links and complementary measurement devices along this path (see Section 2.2). The transmitters and receivers of the links

are both situated on buildings, with the building of the transmitter standing on a small hill (~50 m a.s.l.). The transmitter was

installed at 27 m from ground level (62 m a.s.l.), whereas the receiver is situated 40 m from ground level (51 m above sea

level). The dataset covers the period from 20-08-2014 to 03-01-2016.

2.2. Basic datasets

0 The dataset by VL18 includes records from the following instruments:

- Precipitation categories data classified by five OTT Parsivel laser disdrometers along the microwave link path, at 30 s

intervals (Figure 1). Because of the high temporal resolution, this study only uses the three disdrometers that are located

directly along the link path, so that precipitation falling outside of the link path is not taken into account. The precipitation

categorisation was performed based on the precipitation intensity, hydrometeor size distribution, and hydrometeor fall velocity.

3



- 95 For the purposes of this study disdrometer data were solely used to inquire whether any precipitation was detected along the link path so that this precipitation was not misclassified as fog.
  - A Scintec BLS900 boundary-layer scintillometer, which operates at a frequency of 340 THz (880 nm) and records its signal strength at a sampling frequency of 20 Hz. Even though the BLS900 operates in the near-infrared (NIR), it is assumed that it can be used as an indicator for visibility along the link path to detect the presence of visibility-affecting aerosols such as fog, smoke, and dust. This assumption is made because it was shown by Ijaz et al. (2012) that fog attenuation is not dependent on wavelength within the range of visible and NIR light. Smoke attenuation, however, decreases linearly towards NIR wavelengths. Because in this study this data will be used to detect fog along the link path, this is a favourable condition to be able to use the BLS900 to estimate visibility.
  - A Nokia Flexihopper (38 GHz) microwave link that used to be part of the commercial T-Mobile (currently Odido) cellular network. The Nokia link has a frequency of 38.17625 GHz and a bandwidth of 0.9 MHz. Data are sampled at 20 Hz.
  - For specific periods, time-lapse camera footage was available of both the transmitters and receivers as well as along the link path. Most of these time-lapse images were made during daytime, however, some night-time footage does exist (VL18). This footage was used for a qualitative evaluation of the reference dataset.
- We utilize one-minute humidity data from the university weather station at the Veenkampen in Wageningen (51.981°N,
   5.622°E, henceforth VK). The VK is a fully automated weather station located in a field outside Wageningen approximately
   km west of the link path (Cabrera et al., 2021). The VK also reports visibility, but since fog can be a particularly local phenomenon both horizontally and vertically (van der Meulen, 2007) it was decided that the BLS900 link could provide the most representative visibility data.


Figure 1: A map of Wageningen showing the path of the links in red. The receiving antennas are at the end labelled "Forum"; the transmitting antennas are positioned at the end labelled "Biotechnion". The positions of the disdrometers are indicated with yellow dots. Each dotted position houses one disdrometer, except at the "Forum" position, where two disdrometers and an additional tipping bucket rain gauge are placed. (b) The transmitting antenna mast located on the roof of the "Biotechnion" building. From top to bottom: Scintec BLS900, Nokia Flexihopper (38 GHz) and RAL 26 GHz. The RAL 38 GHz is placed behind the RAL 26 GHz in the photo's perspective and thus not visible. (c) A Parsivel disdrometer (on the "Biotechnion" site). (d) Précis Méchanique tipping bucket rain gauge at the "Forum" site. Image: Van Leth et al. (2018).

Using the NIR BLS900 observations as a proxy for visibility means that we cannot use the commonly used definition for fog of a visibility of 1 km or less, even if it is assumed that the attenuation of the BLS900 is exactly the same as that of visible light. Instead, to create a *fog reference data set* the following method was used: fog is assumed to have occurred if:

- the 30 s mean signal strength received by the BLS900 link dropped below its 15th percentile,
- no precipitation was detected by any of the disdrometers along the link path,
- a relative humidity of 90% or higher was detected at the VK.

The humidity data was included in an attempt to primarily include moments of poor visibility resulting from fog events, whilst excluding as many moments of poor visibility resulting from particulate matter as possible. Note that VK also records black carbon data. It was assumed that this black carbon would most likely represent a background value and would not include the likely much higher incidental black carbon concentrations caused by sources within Wageningen itself. The 15th percentile was chosen because there is a clear signal strength divide in the BLS900 data for what is assumed to be instances of high and low visibility around this percentile of the data. Moreover, this divide provides any ML algorithm with ample training data for the minority class.

Because this study had no direct visibility dataset available, the reference dataset needed to be validated. The primary resource used for this was the time-lapse camera footage from VL18. A number of cases of interest where fog or rain was apparent from the time-lapse camera footage were compared to the fog reference dataset, to check whether it is accurate. While this was not comprehensive, it is nonetheless assumed to be representative of the dataset as a whole.

## 2.4 ML and reference datasets




#### 2.4.1 AutoML deep learning architectures using the McFly package.

To study the potential accuracy ML can provide in the detection of fog from microwave links, as a point of departure the maximum amount of data available (600 data points) for each 30-second timestep from the RSL data was considered. Although this amount of data is not commonly recorded for commercial data links, it is technically possible to do this (Graf et al., 2020). Therefore, it can provide us with the best insight into what could be possible with the current cellular infrastructure.

As the data for each time step represents a densely sampled time series of measured attenuation, the eScience Center's McFly AutoML Python package - specialized for time series classification - was used to identify a suitable ML architecture. Specifically, the primary neural network used was obtained by allowing McFly to train and compare five autogenerated neural networks on a subset of 500 time steps, drawn from the aforementioned binary fog reference data set. In this case, the initial input data was a min-max [0-1] normalised version of the raw RSL data divided in 30-second time series. Because the number of non-fog data points is far greater than the number of fog data points, there are clear majority and minority classes, i.e. the dataset is imbalanced. Accordingly, accuracy makes for a poor evaluation metric for the quality of the model as it would bias the model towards the majority class. Instead, following common practice, the harmonic mean between precision and recall,

https://doi.org/10.5194/egusphere-2025-2634 Preprint. Discussion started: 11 November 2025

© Author(s) 2025. CC BY 4.0 License.

also known as the F1-score, was used because it considers both precision and recall inherently addresses the imbalance reducing the models bias towards the majority class, i.e. making it is much less likely for the model to overestimate the occurrence of the majority class.

## 2.4.2. Input datasets




The data available from the study by VL18 includes timestamped RSL data with a sampling frequency of 20 Hz. This includes the Nokia link that used to be part of a commercial cellular network. Because this link was part of a commercial network, this study will consider it to be the most representative of a standard microwave link. Therefore, the RSL data from the Nokia link was selected to be the main input for the McFly model.

To promote the fog classification, McFly can be provided with additional data channels. The inclusion of two additional predictors was explored. The larger trend of the RSL data and the variable time were hypothesised to be additional predictors of fog occurrence, and thus good additional input data for McFly. Unfortunately, training McFly on separate 30-second time series blocks ignores the time evolution of the meteorological variables. This is unnatural as radiation fog forms and later dissipates in a very characteristic way over the span of minutes to hours: first condensing near the ground as the air cools to the dew point during the night, forming thin ground fog; then growth of the fog layer can occur forming a deep well-mixed fog layer; finally, as the sun rises the fog gradually dissipates from the ground (Price, 2019). This relatively slow development was hypothesised to be a distinguishing property of fog, as opposed to precipitation, which cannot be accounted for in a single isolated 30s interval. To remedy this, a trend variable was created to serve as an additional predictor. This trend variable was chosen to be the rolling rate of change of the RSL over 15 minutes centred on each time step. This includes a wider time frame without having to include the large amount of raw data this covers in the neural network.

Because the fog climatology has a typical annual and diurnal cycle (Román-Cascon et al., 2016; Izett et al., 2019), time and season were hypothesized to be good additional predictors of fog occurrence. To make time a continuous and cyclical variable, both the day of the year and the minute of the day were normalized between -1 and 1. In addition, to incorporate the time and/or trend data of the Nokia signal as input data for the neural network, these predictors were incorporated as additional channels. Because all channels must be the same size, the time and trend variables were padded with copies of themselves to match the size of the RSL channel (600).

## 2.4.3. Experiments

To assess the impact of the different predictors, eight experiments were performed. The full dataset was randomly divided into three fractions: training (70%), validation (15%), and test (15%). As detailed in 2.4.1, the McFly package was used to autogenerate and train five neural networks on 500 random timesteps from the training data with each of the eight input data combinations:

• Random array (1 channel, baseline)

- Only the RSL data (1 channel)
- Only the RSL trend data (1 channel)
- Only the time data (4 channels)
- RSL trend + time data (5 channels)
  - RSL + time data (5 channels)
  - RSL + RSL trend (2 channels)
  - All data (6 channels)

The performance of the resulting models was then judged based on their validation F1-score. The best-performing model from each experiment was then trained on the full training dataset and again validated on the full validation dataset.

The ML technique that performed best according to the McFly package was an Inception Time architecture (see table 1). This type of architecture is derived from the Inception-v4 architecture. It was originally designed for image classification but was later adapted for time series classification. Its defining feature is the parallel convolutions with different kernel sizes (Figure 2). Additional information on the Inception Time architecture and its implementation by McFly can be found in Fawaz et al. (2020) and Van Kuppevelt et al. (2020), respectively.

In the following, the inception time architecture generated by the McFly package and trained on the full data set (respecting the train/test/validation split) is referred to as the McFly model for ease of reference. The fully trained model is available in the 4TU.ResearchData repository (see link in Data Availability Statement).

## InceptionTime



Figure 2: Schematic representation of the architecture of the Inception Time networks generated by McFly according to Van Kuppevelt et al. (2020).

While the ability of the McFly model to match the reference dataset can be expressed in a number of metrics, such as its accuracy, or F1-score, these give little insight into the types of situations where it performs well and where it performs poorly. To remedy this, the model output was compared to the reference dataset for a few interesting individual cases. To give more insight into the local atmospheric conditions during some individual cases the available time-lapse camera footage was used. Also, humidity and temperature data from VK were used as additional verification sources. Altogether this provides an




additional expert judgement about the model's quality. The cases themselves include those discussed for the reference dataset as well as some additional ones and are further discussed in Section 3.

#### 2.5. Fully connected model Wageningen

Unfortunately, in operational cellular communication networks, RSL data is rarely recorded at the frequency of the Nokia link in Wageningen. Additionally, commercial cell towers are generally not equipped with a NIR scintillometer. Therefore it appeared impossible to use the extensive Inception Time model for other datasets. Instead, the outcome of the Inception Time model was used as a reference precision for a reduced model that uses a lower resolution, and a more realistic and readily available array of input variables.

RSL data was available from almost 3000 Nokia and NEC microwave links in a cellular telecommunication network in the Netherlands operated by T-Mobile NL from between 30-05-2012 and 01-09-2012 (Van Leth et al., 2018b). This dataset contained the maximum and minimum power received across each link over intervals of 15 minutes. Additionally, timestamps, frequency, coordinates, and path length were available for each link. For this data, no feasible fog reference dataset could be created. Therefore, the reduced model had to be created and tested around the data from the single link in Wageningen. To do this, the original Wageningen dataset was down sampled to match the time resolution of the T-Mobile dataset, taking the normalized minimum and maximum for every 15 minutes. The four time variables were calculated the same way as before for each time step. The trend variable was calculated similarly to before, but now as a rolling average of the change between the average of the minimum and maximum RSL data over five timesteps, half an hour before and after each time step. This leaves seven input variables as opposed to 3600.

The reduced array of input variables is unsuited for the type of architectures McFly produces since the McFly architectures are designed to detect patterns in one-dimensional time series (Van Kuppevelt et al., 2020). For this reason, it was decided to use a fully connected neural network instead.

In this case, an experiment was also performed, comparing an eight-layer (7 nodes per layer, not tapered) fully connected network utilizing a sigmoid activation function to a network without hidden layers and a sigmoid activation function. This is equivalent to a logistic regression. Both configurations were evaluated using the F1-score metric. The objective of this experiment was to gauge the predictive capability of a basic neural network in contrast to the more intricate Inception Time model and the simpler logistic regression. To properly compare the performance of the reduced model to the McFly model, this study compares its performance not only in the performance metrics but also in the cases discussed for the McFly model.

#### 3. Results




#### 3.1. Reference dataset validation

As no direct secondary fog observations were available to use as a reference dataset, proxy data was used based on relative humidity, disdrometer data, and NIR attenuation. To validate this dataset it was tested against qualitative data points based on direct video observations made with the time-lapse cameras along the link path. A number of cases of interest, where fog or rain was apparent from the time-lapse camera footage were compared to the fog reference dataset, to check whether it is accurate.

#### 255 3.1.1: Case 1: Radiation fog and precipitation

One good case for comparison is 23 August 2014, which had a variation of foggy, clear, and precipitation conditions. Figure 3 shows that until approximately 08:00 CEST the reference data indicates the presence of fog along the link path. This is corroborated by the link path footage, where fog is clearly visible during this time. On top of the mainly clear sky visible in the time-lapse camera footage before and after, the strongly diurnal temperature evolution and quick dissipation of the fog after sunrise suggest that both this and the next day's fog event are cases of radiation fog. Later that day, showers were observed within the link path through this footage. These showers are also visible as short dips in the NIR signal strength of varying intensity. Crucially, these dips in the signal strength are not misidentified as fog in this reference dataset.

Another interesting case is the evening of 25-26 August 2014. The NIR attenuation clearly indicates poor visibility that night and morning, comparable to the two preceding days before but less constant. The disdrometers indicate a long period of precipitation during this time, but the NIR attenuation is far longer-lasting than that of other precipitation events. From the link path footage of 21:00 CEST, it is clear that visibility is indeed relatively poor. The relatively high temperature and near-immediate returning visibility suggest that this is far more likely to be a form of drizzle than precipitation fog.

Overall, smooth and steady NIR attenuation increases like those seen on the mornings of the 23<sup>rd</sup> and 24<sup>th</sup> and seem to be indicative of a clear-cut fog event, while precipitation seems to result in a far more erratic attenuation. The next section will show this observation will largely hold true for the microwave signal as well.

Figure 3: Time series containing several fog and rain events from 22-27 August 2014 CEST. (a) Signal attenuation NIR Scintec BLS900 (solid black line) and its 15<sup>th</sup> percentile (dashed grey line); Veenkampen humidity (solid blue line), and 80% value (dashed blue line); Veenkampen temperature dry vent (solid green line); Disdrometer binary precipitation reference (blue bar). Fog reference dataset (grey bar); Time-lapse image time stamp (grey vertical lines). (b-e) Images from the time-lapse camera mounted near the transmitting antennas on the Biotechnion building aimed along the link path. (b) 23-08-2014 07:00 CEST presumed radiation fog. (c) 23-08-2014 15:00 CEST clear weather. (d) 24-08-2014 12:53 CEST precipitation (shower). (e) 25-08-2014 21:00 CEST precipitation (presumably drizzle).

#### 3.1.2: Case 2: Precipitation fog





Not all fog cases are as clear-cut as the ones described in section 3.1.1. During 16-17 Oct 2015 there was an event where, according to the reference dataset, rain seamlessly transitioned into fog. Figure 4 shows that, although the distinction of the NIR attenuation response holds up to some extent, the difference here is not so clear. Unfortunately, the distinction cannot be made using the time-lapse camera footage. While footage of the rain part of the event was available (Figure 4), the part of the event flagged as fog occurred at night and no footage is available. However, a possible explanation can be gathered from observing the temperature and humidity data. The temperature during the event is quite low, while the humidity only starts to rise as it starts raining. It presumably reaches saturation around 17:00 CEST as it reaches the same level as during the fog that follows. From this and the time-lapse camera footage, it can be concluded that fog already occurred during the rain event. Most likely, the fog event that occurred is an instance of precipitation fog. In other words, fog-wise there might not be a distinction between the rain part and the "fog" part of the event. Because fog, for this study, was defined as "not occurring during rain" to rule out precipitation-caused reductions of visibility, the fog that occurred during the rain event is not included in the reference dataset.

Figure 4: (a) as Figure 3a, but for 16-18 October 2015 CEST. (b) Image from the time-lapse camera mounted near the transmitting antennas on the Biotechnion building aimed along the link path. 16-10-2015 18:00 presumed precipitation fog.

## 295 3.1.3: Case 3: Dense fog


An especially persistent fog event occurred from 11-10-2014 until the morning of 12-10-2014 (Figure 5a). The time-lapse camera footage and reference dataset seem to be in agreement that this fog event lasts from around 18:30 CEST until almost 12:00 CEST the next day. Figure 5b shows that just before the fog forms, a layer of clouds is present above the link path. This makes the formation of fog through radiative cooling unlikely. No further rain is indicated to be present since then either. This could be an instance of advection fog. The layer of clouds above the fog could explain the slow dissipation of the fog layer the next day. However, the temperature profile sheds doubt on this assessment as the fog still seems to be triggered by cooling after sunset.

Figure 5: a) As Figure 3a but for 11-13 October 2014 CEST. (b, c) Images from the time-lapse camera mounted near the receiving antennas on the Forum building aimed along the link path. (b) 11-10-2014 18:00 CEST cloud cover. (c) 12-10-2014 11:00 CEST possible advection fog.

#### 3.1.4: Case 4: Snow and low-temperature fog

Rain is not the only type of precipitation that occurred during the measurement period. A few snow events were also observed. Among them is one during the night and early morning of 24-01-2015. Unfortunately, the snow was not observed directly by the time-lapse camera. However, once the camera clears at 08:30 CEST (Figure 6c) it is clear that it has indeed snowed. It is also apparent that a light fog is present along the link path. This is nicely reflected in the reference data shown in Figure 6a. The snow event in the reference dataset does not distinctly show up in the NIR attenuation. A possible explanation is that fog is already present throughout the event and overshadows any attenuation from the snow similar to the event observed during the precipitation fog. It is impossible to check this as there was no footage from the time lapse cameras available.

Unfortunately, the reference dataset cannot be entirely validated by hand using the time-lapse camera footage. This is both because of the length of the data set and because of the fragmentary coverage of the time-lapse camera footage. Future studies on this subject could therefore greatly benefit from a purpose-made reference dataset using a visibility sensor in or near the link path, ideally near ground level.

Figure 6: (a) As Figure 3a but for the snow event spanning from the 23-25 January 2015. (b, c) Images from the time-lapse camera mounted near the receiving antennas on the Forum building aimed along the link path. (b) 23-01-2015 09:00 CET cold fog. (c) 24-01-2015 08:30 CET cold fog after snow.

#### 3.2. McFly model results Wageningen



From the validation, precision, recall, and specificity in Table 1, it is clear that the baseline model always predicts the majority class (no fog). Even though this model's only input variable is a random array, the imbalance of the dataset ensures that the accuracy is high. The F1-score, on the contrary, provides the more accurate assessment with a value of zero. These values provide a baseline for all other models to be tested against.

Interestingly, whilst a model trained on RSL data performs better than the baseline, the models trained on RSL trend data, the time date, and their combination do not. However, with the exception of the RSL + RSL trend combination, a model trained on RSL in combination with any of these other channels performs better than a model that is only trained on the RSL data. While adding additional channels is expected to generally increase the accuracy of a deep learning model, it might be the case that in this instance it results in overfitting on the training dataset, resulting in a lower performance on the validation dataset. Conceptually, it makes sense that time and the RSL trend are bad predictors on their own. If time of day and time of year are indeed assumed to be good predictors for fog, then fog would need to be so common during certain parts of the year and/or day that fog data points during these periods outnumber non-fog data points for it to be a good predictor on its own. The RSL trend, on the other hand, looks at a rolling rate of change over 15 minutes. Without the context of the local RSL data, this trend



can only detect a change within a half-hour period. A model based solely on this data could not distinguish between a constant attenuation lasting longer than 15 minutes and no attenuation at all. Additionally, it would detect a singular change for 15 minutes.

Table 1 illustrates that the model exhibiting the maximal F1-score is the one that utilized all accessible data channels. Therefore, it is this model that is used to explore the potential of fog detection using microwave link signal strength. The other metrics provide insights into the model's strengths and weaknesses. The high accuracy and specificity are not surprising. They are the result of the imbalance in the dataset. The large negative majority class in the dataset ensures that any model that predicts a large number of negatives will be correct most of the time. Precision and recall are much more informative here. The relatively high precision indicates that the model does not misidentify many instances of non-fog as fog. The model performs worse in recall, which means that many instances of fog are not properly identified as such.

Table 1: Performance metrics of McFly networks for different input data and the architectures used. Appendix A summarizes the meaning of the metrics.

| Experiment              | Architecture   | Number in<br>4TU data<br>repository | Precision | Recall | Specificity | Accuracy | F1-score |
|-------------------------|----------------|-------------------------------------|-----------|--------|-------------|----------|----------|
| Random array (baseline) | -              | 0                                   | 0.000     | 0.000  | 1.000       | 0.912    | 0.000    |
| RSL                     | ResNet         | 1                                   | 0.800     | 0.247  | 0.994       | 0.929    | 0.377    |
| RSL trend               | Deep ConvLSTM  | 2                                   | 0.000     | 0.000  | 1.000       | 0.912    | 0.000    |
| time                    | ResNet         | 3                                   | 0.000     | 0.000  | 1.000       | 0.912    | 0.000    |
| RSL trend + time        | ResNet         | 4                                   | 0.000     | 0.000  | 1.000       | 0.912    | 0.000    |
| RSL + time              | ResNet         | 5                                   | 0.636     | 0.469  | 0.974       | 0.930    | 0.540    |
| RSL + RSL trend         | Inception Time | 6                                   | 0.624     | 0.132  | 0.992       | 0.917    | 0.218    |
| all data                | Inception Time | 7                                   | 0.717     | 0.464  | 0.982       | 0.937    | 0.564    |

Using the aforementioned cases as well as some new ones, the model output was compared to the reference dataset. This provides a more intuitive assessment of the model's quality.

#### 355 3.2.1: Case 1: Radiation fog and precipitation

Figure 7 shows that the radiation fog events of the mornings of 23 and 24 August 2014 are accurately detected by the Inception Time model. However, the model seems to consistently 'detect' the occurrence of radiation fog about half an hour before the reference data indicates fog actually occurred. It is unclear why this is the case.


The Inception Time model does not misidentify the afternoon showers as fog either. Instead, where the model seems to struggle is the longer-lasting drizzle event from the evening of 25-08-2014 until the morning of 26-08-2014. It misidentifies this rain event as fog between 22:00 and 08:00 CEST. The mean Nokia attenuation is similar in magnitude for the drizzle event as during the two radiation fog events in that figure. This suggests that this model does not pick up on the fluctuations in this precipitation event as being more distinctive to rain than to fog. Notably, not the entire rain event is flagged as fog. Instead, the fog classification seems to be given only during the night and early morning hours.

Figure 7: Time series containing several fog and rain events spanning from the 22nd until the 27th of August 2014 CEST. (a) Signal attenuation NIR Scintec BLS900 (solid black line). 30 s mean signal attenuation Nokia Flexihopper microwave link (solid red line), and 15-min min and max (solid pink lines). Disdrometer binary precipitation reference (blue bar). Fog reference dataset (grey bar). McFly Inception Time fog identification (red bar). Fully connected model fog identification (green bar). Logistic regression fog identification (yellow bar). Time-lapse image time stamp (grey vertical lines). (b-e) Images from the time-lapse camera mounted near the transmitting antennas on the Biotechnion building aimed along the link path. (b) 23-08-2014 07:00 presumed radiation fog. (c) 23-08-2014 15:00 CEST clear weather. (d) 24-08-2014 12:53 CEST precipitation (shower). (e) 25-08-2014 21:00 CEST precipitation (presumably drizzle).

The same seems to have occurred during other nighttime rain events. The short rain events between 4:00 CEST and 6:00 CEST on 30-08-2014 (Figure 8) are marked as fog by the model. Very similar events happen only hours later but are not misidentified as fog. This suggests the model marks a sufficiently large drop in the signal strength as fog as long as it happens during the right time window, while it ignores anything outside this time window. It appears that the model relies heavily on the time variables that are part of the input.




Figure 8: Time series containing a fog and several rain events spanning from the 28th until the 31st of August 2015 CEST. Signal attenuation NIR Scintec BLS900 (solid black line). 30 s mean signal attenuation Nokia Flexihopper microwave link (solid red line), and 15-min min and max (solid pink lines). Disdrometer binary precipitation reference (blue bar). Fog reference dataset (grey bar). McFly Inception Time fog identification (red bar). Fully connected model fog identification (green bar). Logistical regression fog identification (yellow bar). Time-lapse image time stamp (grey vertical lines).

While an overreliance on the time variables is not ideal, the model does use the time data to great effect. Time might not be directly correlated with fog, but it is a simple indicator that is always available for any location. This makes it far more useful than humidity or temperature data, which would need dedicated sensors and would somewhat defeat the premise of using the existing infrastructure. Moreover, the precipitation being misinterpreted as fog does not seem to be the most prominent issue, as according to the precision, the model is relatively good at avoiding false positives. Given that the difference between fog and precipitation is readily apparent from the erratic behaviour of the Nokia RSL during precipitation as opposed to the smooth signal during fog events, a future improved model might make use of other inputs that consider a longer period of time such as the variance, or increase the time step size so that longer scale variations can be taken into account.

#### 3.2.2: Case 2: Precipitation fog

Far more common than false positives like those in the case of the drizzle and rain events are the false negatives like those that occurred on 17-10-2015 (Figure 9a). What is assumed to be a precipitation fog event is not recognised as fog by the model. The only minute part of the event that was flagged as fog occurred during the precipitation event that preceded the fog event in the reference dataset. It would be intuitive for the entire event to be flagged as fog, including the precipitation, as that is what the NIR attenuation and time-lapse would suggest.

However, unlike from the NIR signal, the instances of precipitation are very apparent from the Nokia signal. These are the instances of fog that are not apparent from the attenuation of the Nokia signal. It appears the fog in these instances causes little to no attenuation in the Nokia signal despite a haze being very visible in the time-lapse footage in Figure 9b and the NIR signal being completely blocked. Therefore it is not surprising that that these instances of fog were not detected by the model.




Figure 9: Time series containing a fog/rain event spanning from the 16-18 October 2015 CEST. (a) Signal attenuation NIR Scintec BLS900 (solid black line). 30 s mean signal attenuation Nokia Flexihopper microwave link (solid red line), and 15-min min and max (solid pink lines). Disdrometer binary precipitation reference (blue bar). Fog reference dataset (grey bar). McFly Inception Time fog identification (red bar). Fully connected model fog identification (green bar). Logistic regression fog identification (yellow bar). Time-lapse image time stamp (grey vertical lines). (b) Image from the time-lapse camera mounted near the transmitting antennas on the Biotechnion building aimed along the link path. 16-10-2015 18:00 CEST presumed precipitation fog.

It seems that the properties of precipitation fog make it significantly less attenuating in the microwave range of the Nokia link. While it falls outside of the scope of this research, a possible explanation could be that the droplets in precipitation fog are an order of magnitude smaller than those in radiation or advection fog. According to Rayleigh-theory, the microwave attenuation in this frequency range is roughly linearly proportional to droplet volume at the range of droplet sizes observed in fog (Pendleton, et al., 1994). Meanwhile, NIR has a peak in its extinction spectrum at droplets of around 2 μm, a common size in precipitation fog (Niu et al., 2012). Precipitation fog seems to have significantly smaller droplets than other types of fog. The extinction coefficient at this droplet size is around twice that of larger fog droplets (Perelet et al., 2021). This makes precipitation fog disproportionally attenuating for the BLS900, while it is very hard to detect with microwave links. This does not directly translate to visibility since visible light, e.g. 550 nm, has its strongest extinction peak at smaller droplet sizes (0.5 μm, Bernardin et al., 2010).


## 3.2.3: Case 3: Dense fog

A more complex yet not dissimilar issue seems to be the case for the long-lasting fog event from the evening of 11-10-2014 till the morning of 12-10-2014 (Figure 10). The core of the fog event is recognised as fog by the model. However, unlike in the case of radiation fog, the detection seems to be an hour late. Moreover, even though fog is clearly present in the time-lapse footage and is thick enough at 11:00 CEST to obscure everything except for the roof edge from the time-lapse camera, the model does not recognise this as fog. Similar to the case of precipitation fog, the mean Nokia attenuation at this time does not seem to be higher than before the event. This is in contrast to that of the NIR attenuation.

The Nokia attenuation varies greatly during the event. Up until around 00:00 CEST the Nokia attenuation seems to behave quite typically compared to radiation fog, steadily increasing until it reaches a peak. However, after 00:00 CEST there is a steep drop that correlates with a dip in the NIR attenuation. While the NIR attenuation then rises again the Nokia attenuation continues declining, albeit slowly, accelerating a little around 06:00 CEST, until it reaches a similar attenuation as before the event. This again explains why the model is unable to discern the presence of fog from the Nokia attenuation.

Figure 10: Time series of a fog event spanning from the 11th until the 13th of October 2014 CEST. (a) Signal attenuation NIR Scintec BLS900 (solid black line). 30 s mean signal attenuation Nokia Flexihopper microwave link (solid red line), and 15-min min and max (solid pink lines). Disdrometer binary precipitation reference (blue bar). Fog reference dataset (grey bar). McFly Inception Time fog identification (red bar). Fully connected model fog identification (green bar). Logistic regression fog identification (yellow bar). Time-lapse image time stamp (grey vertical lines). (b, c) Images from the time-lapse camera mounted near the receiving antennas on the Forum building aimed along the link path. (b) 11-10-2014 18:00 CEST cloud cover. (c) 12-10-2014 11:00 CEST possible advection fog.

### 3.2.4: Case 4:Snow and low-temperature fog

As mentioned previously, some snow was also observed during the measurement period. Fortunately, this snow fell during the night and early morning and can therefore be compared to fog events if time is indeed a dominant factor in distinguishing fog from precipitation. Figure 10a shows that not all nightly disturbances in the Nokia RSL are flagged as fog. In this precipitation event, the majority of snowfall was correctly distinguished from the fog that occurred after.

Figure 11: Time series containing several fog events and a snow event spanning from the 23<sup>rd</sup> until the 25<sup>th</sup> of January 2015 CET.

(a) Signal attenuation NIR Scintec BLS900 (solid black line). 30 s mean signal attenuation Nokia Flexihopper microwave link (solid red line), and 15 min min and max (solid pink lines). Disdrometer binary precipitation reference (blue bar). Fog reference dataset (grey bar). McFly Inception Time fog identification (red bar). Fully connected model fog identification (green bar). Logistical regression fog identification (yellow bar). Time-lapse image time stamp (grey vertical lines). (b, c) Images from the time-lapse camera mounted near the receiving antennas on the Forum building aimed along the link path. (b) 23-01-2015 09:00 cold fog. (c) 24-01-2015 08:30 cold fog after snow.

Regrettably, fog occurring later in the morning was not properly identified. Surprisingly, the fog event from the previous night was not identified either, despite fully obscuring the BLS900. The cause of this is evident from the Nokia RSL. Throughout the entirety of the event, there appears to be negligible signal attenuation, except for a minor, inconsequential anomaly observed around 03:00 CET, despite clear evidence of fog captured in the time-lapse camera footage. This discrepancy is likely attributable to temperature variations. Notably, during this period, ambient temperatures recorded at VK were below freezing (Figure 11a). It is relevant to consider that measurements at VK were conducted near ground level outside city limits, while the link path resided approximately 30 m above ground level within city limits. Consequently, the precise temperature along the link path likely deviates from the VK measurements. This lack of attenuation was consistent with other low-temperature




fog events. At these temperatures, it is unlikely that the fog consists of ice crystals (Gultepe et al., 2017). Although the low dielectric constant of ice in the microwave range would explain the lack of attenuation in the Nokia signal (Battan, 1973), while the dielectric constant of water does change as its temperature decreases (Andryieuski et al., 2015). However, this is unlikely to be sufficient to make the attenuation by the fog droplets negligible compared to the background attenuation, as even a temperature difference this small would only affect the dielectric permittivity by a few percent at 38 GHz (Andryieuski et al., 2015).

Droplet size might also explain the lack of microwave attenuation during low-temperature fog events. At lower temperatures, the saturation vapour pressure decreases as well. According to Köhler theory, this leads to an increase in the relative humidity of the air for a fixed amount of water vapor, which favours the formation of fog droplets. As a result, smaller droplets can form at lower temperatures, as the air becomes more supersaturated with respect to water vapor. This might result in a droplet size distribution at the observed temperatures that does not cause enough attenuation in the Nokia link to detect fog.

#### 3.3. Fully connected model Wageningen

It is important to note that both the fully connected model and the logistic regression model produce a 30 times lower temporal resolution output than the McFly Inception Time model. While this complicates a direct comparison with the McFly model, some inductions can still be made. Both the fully connected model and the logistic regression model show a similar pattern as the Inception Time model. Table 2 indicates that specificity, and subsequently accuracy, are high, as expected, while precision and recall are lower. Precision is substantially higher than recall for both models. This difference is even bigger than for the Inception Time model. Precision is higher while recall is lower. This indicates that both the reduced models go for the "low-hanging fruit", as it were. They seem to mark a lower fraction of time steps as fog. The ones they mark as fog are more often true positives, but as a result, both models produce a higher fraction of false negatives. Overall, the F1-score of the logistic regression is lower than that of the fully connected model, and both are lower than that of the Inception Time model. This pattern is also clear when looking at some of the aforementioned cases.

Table 2: Performance metrics of Fully connected and Logistic regression networks compared to McFly Inception Time network including resolution.

| Model                      | Resolution | Precision | Recall | Specificity | Accuracy | F1-score |
|----------------------------|------------|-----------|--------|-------------|----------|----------|
| Inception Time (Sect. 2.4) | 30 s       | 0.717     | 0.464  | 0.982       | 0.937    | 0.564    |
| Fully Connected (Sect 2.5) | 900 s      | 0.825     | 0.322  | 0.993       | 0.935    | 0.463    |
| Logistic Regression        | 900 s      | 0.784     | 0.208  | 0.995       | 0.926    | 0.328    |





## 3.3.1: Case 1: Radiation fog and precipitation

Figure 7a again shows the typical fog events of the mornings of 23 and 24 August 2014. It is clear that both the fully connected model and the logistic regression model recognise the attenuation on the morning of 23-08-2014 as a fog event. However, both seem to leave out a few time steps in this fog event, where the Nokia attenuation is no longer increasing. The Logistic Regression model only recognises the fog after the attenuation reaches a certain strength. This is probably also the reason that it does not recognise the fog event of 24-08-2014. This makes sense for a logistic model, as these types of models base their output directly on the absolute values of the input parameters. The fully connected model does recognise this fog event, but it also leaves out some time steps. This might be an interaction between how the trend variable is defined and the relatively short length of the fog event. In future parsimonious models, it might therefore be advantageous to include the data from a number of timesteps surrounding each time step as input data in addition to, or instead of, a calculated trend. Neither of the models misidentify the short fog events during this period as rain despite the attenuation being particularly high. However, the fully connected model does make the same misidentification as the McFly model for the longer-lasting drizzle event from the evening of 25-08-2014 till the morning of 26-08-2014, similarly misidentifying this rain event as fog between 22:00 CEST and 08:00 CEST. Seemingly, it makes the same time distinction here, although here its 'detection' is again less consistent. The Logistic Regression model does not indicate this rain event as fog at all, apart from the single time step with the highest attenuation. This lends credibility to the idea that the Logistic Regression model only flags time steps as fog if the attenuation hits a certain threshold.

Neither one of the parsimonious models misidentifies any of the rain events from 30-08-2014, in Figure 7a, as fog despite the min and max attenuation for these events being significantly higher than that of the drizzle event or that of the fog event on the day before. Since the fog event of the day is mostly accurately identified by the fully connected model as opposed to the Logistic Regression model, and it occurs around the same time, it implies that the trend variable is successfully employed by this model to differentiate these rain events from fog. In this case, it does so more successfully than the Inception Time model. A possible reason for this is that the trend variable of the smaller models looks at a longer period of time, which would make it easier to disregard relatively short-lasting rain peaks.

#### 3.3.2: Case 2: Precipitation fog

As was earlier concluded, the failure of the Inception Time model to correctly identify precipitation fog was the result of this type of fog not creating enough attenuation in the spectrum of the Nokia Flexihopper to be identified at all. Therefore, it is not surprising that neither of the reduced models could do it either (Figure 9a). While inconsequential, it is notable that the fully connected model indicates the presence of fog around the same time during the preceding rain event as the Inception Time model.



#### **3.3.3:** Case **3:** Dense fog

As seen in Figures 7a and 8a, the reduced models, and especially the fully connected model, were inconsistent in identifying fog for radiation fog events, often correctly identifying events but leaving out several time steps during the event. This is not the case for the dense fog event of 11-12 October 2014 (Figure 10). Here, both models make a solid prediction for the core of the event, similar to the Inception Time model. Unsurprisingly, a similar misidentification is made to that of the Inception Time model at the latter half of the event during the slow decline of the signal attenuation. Both parsimonious models do start this misidentification about 1.5 hours sooner. More surprisingly, something similar happens at the start of the fog event during the fairly typical increase of the Nokia signal attenuation. Given that the attenuation should be strong enough for both models to pick it up earlier, and the trend variable should be about the same as for the aforementioned radiation fog events, this might be an instance of the opposite time issue as before. Fog occurs much earlier here than in the other cases. These models might have learned to disregard everything outside a certain time frame as non-fog.

## 3.3.4: Case 4: Snow and low-temperature fog

Just like the Inception Time model, both models make a short fog "identification" during the snow event. However, it seems that overall the reduced models do not misidentify snow as fog. As mentioned before, it is unclear whether fog actually occurred during the snow event, but if it did it is unlikely that this is what caused the Nokia attenuation.

Similarly to the precipitation fog case, the Nokia attenuation from the cold fog is not strong enough to be identifiable from the dataset. This is again less of a problem with the models themselves and more with the properties of this microwave link, if not with microwave links as a tool for fog detection in general.

## 3.4. Fully connected model Netherlands

When the fully connected model was provided with the normalised RSL data from the country-wide cellular telecommunication network links, some interesting patterns emerged. For the overwhelming majority of time steps no or almost no fog is identified along any of the link paths. However, for a few time steps a large number of link paths are indicated to have fog in them. Moreover, the links that indicate fog tend to be in close proximity to each other. This is a promising sign as this is what would be expected when real fog is detected along the link path.

To check whether this is indeed the case, the fog identification by the fully connected model is compared to known fog and precipitation events using visibility and precipitation observations by KNMI automated weather stations of De Bilt, Deelen, and Twenthe and spatial precipitation data from the KNMI radar product for the 26<sup>th</sup> and 27<sup>th</sup> of August 2012.


## 3.4.1: Case 1: Fog

According to visibility observations (Figure 12) and recorded during the night and early morning of the 27 August 2012 by the KNMI weather stations of De Bilt, Deelen, and Twenthe, there was a sharp drop in horizontal visibility lasting several hours. For each of the stations, visibility dropped to less than 1000 m while no rain was reported during this time. Assuming this drop in visibility was caused by atmospheric water droplets, this meets the generally accepted definition of fog. These fog conditions were first reported in De Bilt, approximately an hour later in Deelen, and approximately an hour after that in Twenthe (Figure 12). This suggests that the atmospheric structure that caused the fog conditions travelled from west to east over the country.

The fog detections made based on the 10 closest telecommunication links shown below the station visibility data in Figure 12 echo this. While not all of the links seem to detect the fog event and the length of the detection does not match the length of the fog event based on the visibility data, the general structure is detected. The majority of the links give a fog detection signal between 15 minutes and an hour during the fog event within the same hour per station. The fog detection also seems to pick up on the movements of the fog bank from west to east in the same interval as the weather stations.

Figure 12: Three time series of a fog event and several rain events spanning from the 26<sup>th</sup> until 12:00 on the 27<sup>th</sup> of August 2012 CEST. Hourly horizontal visibility (solid black line), and hourly precipitation reference (blue bar) as reported by the KNMI weather stations (a) De Bilt, (b) Deelen, and (c) Twenthe). For each station fog identification, by the fully connected model, for the 10 closest T-Mobile NL cellular telecommunication links ordered by proximity from top to bottom (green bars).

The potential capability of the fully connected model to detect the extent and development of fog events becomes clearer if the fog detection per link is depicted on a map (Figure 13). The quarterly maps clearly show a band of what is presumably a fog bank passing over the country in the span of three hours. While not all links seem to report fog, the pattern is clear.





Figure 13: Spatial maps containing quarterly fog identification, from 00:00 on the 26<sup>th</sup> until 02:45 on the 27<sup>th</sup> of August 2012 CEST, by the fully connected model, based on per link RSL data from the T-Mobile NL (currently Odido) cellular telecommunication network. Each link is represented by a line connecting its transmitter and receiver. A blue line indicates the absence of fog in that link path; a red line indicates the presence of fog in that link path; a grey line indicates the absence of data for that link path. The green location markers from west to east mark the locations of the KNMI weather stations of; De Bilt, Deelen, and Twenthe.

Regrettably, it is unfeasible to pinpoint whether the fully connected model's failure to detect the full duration of this fog event stems from insufficient training data from other sources, such as variations in fog droplet size, fog bank depth, or any other factor, without supplementary data, like time-lapse camera footage or temperature and humidity data with higher temporal resolution. Neither is it possible to investigate more local occurrences of fog or do any meaningful quantitative analysis without a higher-quality reference dataset. Possibly fog detection from satellite observations (Lee et al., 2011) and community science weather data may enrich the data richness in future studies.

#### 3.4.2: Case 2: Precipitation

Regrettably, a large number of the fog instances detected by the fully connected model are most likely the result of precipitation. Figure 14 shows that on the 26 August 2012, many of the nearby links detected fog during hours when the corresponding stations reported rain in the last hour. While it cannot be ruled out that these were instances of precipitation fog, the time of day would suggest this is highly unlikely.

More encouraging is the comparison of a spatial representation of the fully connected model fog detection (Figure 14a) to the radar product (Figure 14b) for 26 August 2012 10:00 CEST. While the correlation between rain and links that detected fog is far from absolute, a similarity between their spatial distributions appears.

That the model has trouble distinguishing fog and rain is not surprising given the low temporal resolution of the attenuation data. That this is more visible with more links that the model did not train on is to be expected. What is surprising is the fact that many of these misidentifications of fog took place during the late morning and afternoon. This was very rare with the Wageningen dataset, and should logically be based on the time data in the model input data. It is therefore unclear why this mistake is so prevalent in this case.

28

Figure 14: Spatial maps comparing fog detection by the fully connected model to precipitation for 26<sup>th</sup> of August 2012 10:00 CEST. (a) Fog identification by the fully connected model, based on quarterly per link RSL data from the T-Mobile NL cellular telecommunication network. Each link is represented by a line connecting its transmitter and receiver. A blue line indicates the absence of fog in that link path; a red line indicates the presence of fog in that link path; a grey line indicates the absence of data for that link path. (b) Precipitation according to 5-minute data from KNMI. (a, b) The green location markers from west to east mark the locations of the KNMI weather stations of; De Bilt, Deelen, and Twenthe.

#### 4. Discussion





#### 4.1. Microwave links as a data source

The largest asset of using microwave links as a data source for fog monitoring is that the infrastructure already exists and has a wide coverage worldwide. However, it does come with some inherent limitations. Obviously the used frequencies are optimised for data transfer instead of fog monitoring. D13b concluded that the use of microwave links in the range of 38 GHz is promising for fog monitoring, especially in cases of heavy fog, which create severe visibility limitations particularly as it drops to the tens of meters range. They also expressed an expectation that using higher-resolution and frequency measurement systems to observe fog would improve the detection and the ability to track lighter fog. However, while these assumptions make sense based on their limited data and case studies, the fog density does not seem to be the limiting factor. While it was not the primary aim of our study, its case studies show that the fog density does not seem to be the primary contributor to the attenuation intensity of a 38 GHz link. E.g. the fog on 12-10-2014 at 11:00 CEST is much denser than any of the other fog events while causing a negligible amount of signal attenuation. This underlines their recommendation for analysing a large number of events in different condition ranges and from different geographical regions. While our study is a first step towards




that, the different behaviours of the signal attenuation for different cases and the lack of conclusive explanations show that there is still much to gain there.

It is still true that microwave links in the studied range cannot detect all fog events, either partly or completely. While the hypothesis that droplet size might be a contributing factor in this large variation of attenuating properties of fog will be discussed later, if it is true, D13b's expectation that higher frequency links, e.g. around 60 or 80 GHz, would be able to improve measurements remains valid.

Another limitation inherent to any fog monitoring using microwave links is their location. While their location can be considered near ground level (tens of meters), a shallow fog layer may be found below that. In these cases, the fog can only be detected after it thickens, detaches, or not at all.

#### 630 4.2. Data availability

The main limitation of this study is that it relies on pre-existing data. While the quality of the data is generally good, it was not collected with fog monitoring in mind. That notwithstanding, the frequency and extent of the data used still exceeds that of any other study on the use of microwave links as a fog monitoring tool.

As mentioned in the methods, no in situ fog reference datasets existed for either the single microwave link in Wageningen or for the country-wide network. One had to be created for Wageningen by combining pre-existing humidity data from the nearby VK station, the Scintec BLS900 scintillometer, and the Parsivel disdrometers. This reference dataset could later be validated for a number of events based on the time-lapse camera footage. However, none of these instruments is intended for fog monitoring. This hampered defining a dataset that was consistent with the commonly used definition of a visibility of 1 km or less.

Concerning the country-wide network, the best reference data available per link was the visibility data of the nearest weather station. Unfortunately, these stations are not close enough to any of the links to use them as a reference dataset for ML. That means that the ML models could only be trained on data from a single link. For a general model that works well for the whole network, said model should be trained on more links jointly taking the spatial extent of the fog phenomenon into account.

As mentioned previously, the temporal resolution of our microwave link data, 15 minutes, is orders of magnitude higher than that used in previous studies (D13a,b, D15), which had a resolution of 24 hours. The Wageningen link, however, had data recorded at a temporal resolution of 50 ms. As this study has shown, this higher resolution is much more suitable for a technique like ML. It allows for the use of much more complex network architectures that are purpose-built for time series recognition. These models have been demonstrated to be better at distinguishing fog from precipitation. Having this temporal resolution available for the whole network would be the logical next step in constructing a functional fog monitoring system based on the existing telecommunication network.

Finally, the country-wide microwave link was collected over a number of summer months, while the fully connected model was trained on Wageningen link data collected over more than a full year. Because fog more often takes place in autumn and winter in The Netherlands, this data is absent from the country-wide dataset. As a result of both datasets being min-max

https://doi.org/10.5194/egusphere-2025-2634 Preprint. Discussion started: 11 November 2025

© Author(s) 2025. CC BY 4.0 License.

normalised, the datasets are not equally balanced. Having the datasets be collected over a similar or longer time span would mitigate this problem substantially.

#### 4.3. Methods



While the McFly model was better than the fully connected model at distinguishing precipitation from fog, it still makes classification mistakes that are readily apparent to a human observer from the attenuation data. A plausible reason for this is that the McFly models as used in this study only look at a period of 30 s per classification time step, and all time steps are assumed to be independent from each other. While a trend variable was provided to the model, this only gave a global trend and did not prove to be a very productive addition to the model input data.

Future studies could consider a number of alternatives in input data or ML methods to improve the performance of ML for fog detection. A future model could use a longer data window per time step. A longer time window might be better at distinguishing the more erratic attenuation fingerprint of precipitation events from the smooth increase of attenuation typical for fog events. This would, however, make the model exponentially slower as it would greatly increase the amount of data it works with. Temporal resolution could be sacrificed for a more favourable balance of window size and data resolution. Alternatively, a future model could be trained on time windows of different lengths with different channels, where the longer time windows would consist of lower-resolution data.

Finally, future studies might explore various deep learning neural networks. McFly uses Convolutional Neural Networks for time series classification. While these models are skilled in finding patterns, they assume all time steps are independent, which does not apply to fog time series. Deep learning neural network types like Recurrent Neural Networks or Long Short-Term Memory Networks (e.g. Pudashine et al., 2020) are specifically designed to deal with this type of dependent data.

### 5. Conclusions

This study used machine leaning model techniques to detect fog within commercial cellular communication networks in order to assess the potential of advancing the observational coverage of fog events. The McFly software package was used to identify which machine learning model was best suited using cellular microwave signal as main input, and additional input from nearby weather station data for Wageningen (The Netherlands) as a test case. The InceptionTime machine learning approach was found to achieve the highest score. The presented results underscore the potential of machine leaning as a valuable tool for discerning fog occurrences from precipitation using microwave telecommunication link signal strength. This advancement marks a significant stride in leveraging these links for large-scale, high-resolution real-time fog monitoring. However, while our method shows promise, this study primarily serves to illustrate the future prospects of ML in fog detection via microwave telecommunication links.

Acknowledgements: We acknowledge funding from the NL eScienceCenter project "Opportunistic Sensing of Hydrometeors with Commercial Microwave Links", under number SSIML-2021-008. We thank our collaborators in this project (Hidde Leijnse, Aart Overeem and Kirien Whan from KNMI; Luuk van der Valk, Bas Walraven and Ruben Imhoff from Delft University of Technology; and Linda Bogerd from Wageningen University) for fruitful discussions on data and methods. We thank Pieter Hazenberg for his contribution to installing and maintaining the instruments and Henk Pietersen for his help in the preparation for the Wageningen measurement campaign. The Nokia Flexihopper link system was kindly provided by T-Mobile Netherlands (currently Odido Netherlands). The OTT Parsivel disdrometers were provided by Alexis Berne and colleagues from the École Polytechique Fédérale de Lausanne (EPFL) in Switzerland. Funding for this campaign was provided by the former Netherlands Technology Foundation STW, currently NWO-TTW (project 11944). We gratefully acknowledge Ronald Kloeg and Ralph Koppelaar (both at T-Mobile NL, currently Odido NL) for providing the cellular communication link data set for The Netherlands. We thank Sjoerd Barten (Wageningen University) for assisting on Figure 12.

#### Data availability

The link and disdrometer data collected in this campaign is made publicly available at the 4TU data repository and can be found at https://doi.org/10.4121/uuid:1dd45123-c732-4390-9fe4-6e09b578d4ff. The automatic weather station data from the Veenkampen in Wageningen are available at <a href="https://maq-observations.nl/">https://maq-observations.nl/</a>. Data from routine KNMI weather data are accessible through <a href="https://www.knmi.nl/kennis-en-datacentrum/uitleg/automatische-weerstations">https://www.knmi.nl/kennis-en-datacentrum/uitleg/automatische-weerstations</a>. The McFly software package is available at <a href="https://github.com/NLeSC/mcfly">https://github.com/NLeSC/mcfly</a>. The machine learning model architectures are available at 4TU.ResearchData (dx.doi.org/10.4121/e3a19ed4-801f-4586-9ba1-340c4fc3b08e).

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

810

800

## Appendix A: Overview of utilized statistical metrics to assess machine learning model scores.

| Metric       | Description                                                                           | Calculation                                                       |  |  |  |
|--------------|---------------------------------------------------------------------------------------|-------------------------------------------------------------------|--|--|--|
| Precision    | Measures the accuracy of nositive predictions                                         | True Positives / (True Positives + False Positives)               |  |  |  |
|              | Measures how well the model identifies all positive instances.                        | True Positives / (True Positives + False Negatives)               |  |  |  |
| ISpecificity | Measures how well the model identifies all negative instances.                        | True Negatives / (True Negatives + False Positives)               |  |  |  |
|              | Measures the overall correctness of the model.                                        | (True Positives + True Negatives) / (Total number of predictions) |  |  |  |
| F1-Score     | A balanced measure of precision and recall, especially useful in imbalanced datasets. | 2 * (Precision * Recall) / (Precision + Recall)                   |  |  |  |