# Peer review of "Fog Monitoring through Machine Learning of Signal Attenuation Data from Microwave Links from Cellular Communication Networks"

_EGUsphere, 2025_

## Referee Comment (RC1)

**General Assessment**

This manuscript presents systematic investigation into the potential of using commercial microwave communication links, first presented David et.al.,  and suggest machine-learning methods to detect fog where rain may be the cause for signal attenuation. The topic is highly relevant for atmospheric monitoring, transportation safety, and applied meteorology.

The work consists of the following steps:
1. Construct a reference data set for training and validation.
2. Construct the appropriate McFly/InceptionTime architecture
3. Train it on high temporal-resolution experimental dataset from Wageningen
4. Apply its reduced version to a national-scale commercial network

The work represents a comprehensive study with many details. However, in terms of novelty, its claimed contribution is in the use of the McFly/InceptionTime architecture for fog detection. Considering the constraints and the limitations of the available data, the results are reasonable and justify publication. However, there are several methodological and interpretational issues that should be addressed before publication.
* * *
**Major Comments**

1. **Reference Dataset Construction.** The paper aims at distinguishing between rain and fog events. However, dry periods can also be identified as fog, especially with high levels of humidity. It is therefore important that the reference dataset consist of rain and fog events, as well as dry periods. Moreover, for meaningful accuracy assessment of any ML-algorithm, the reference dataset should be balanced.

2. **Reference Dataset Validation**. Existing validation of the reference data set is qualitative. A quantitative analysis, indicating on misdetection vs. false identification of fog in rainy/dry periods via an e.g., ROC , is required.

3. **The McFly/InceptionTime architecture.** Since the architecture has been chosen by comparing performance of several options, the performance measure is crucial. As mentioned, imbalanced data leads to optimistic accuracy performance. But, what really counts as a performance measure is the accuracy. Therefore, it is recommended to balance the data, as often done in similar applications[1]. Alternatively, using the imbalanced data, indication on false detection may indicate on the effect of the long no-fog periods.

4. **Wet/Dry classification.** As mentioned, previous studies of Fog and CML excluded rain periods, while this study does not. However, a discussion of the
* * *
[1] See, for example, Habi, Hai Victor, and Hagit Messer. "Recurrent neural network for rain estimation using commercial microwave links." *IEEE Transactions on Geoscience and Remote Sensing* 59.5 (2020): 3672-3681.

possibility of combining wet/dry classification with fog/rain classification is missing.

5. **Misclassification.** Treating 30-second segments as independent samples does not adequately represent the temporal evolution of fog. This likely contributes to misclassification of drizzle and weak precipitation and even high humidity. Consideration of temporal sequence models or multi-scale temporal inputs is strongly recommended.

6. **Generalisation and Dataset Mismatch**
While application of the method of country wide network is important, and is probably the major novelty in the paper, it needs consideration. In particular, the reduced model is trained on a single-link dataset but applied to a national network with different temporal coverage and climatological conditions. The manuscript should more clearly acknowledge and discuss the limitations of this generalisation.

7. **Interpretation of Physical Mechanisms**
Several explanations (e.g., droplet-size effects, Rayleigh attenuation assumptions, cold fog behaviour) are presented without sufficient direct support from the literature. These interpretations should be more cautiously framed and supported with appropriate references.

**Minor Comments**

**1. Construction and validation of the fog reference dataset.** This is the most consequential methodological step, yet the description is somewhat subjective and potentially yields systematic biases. It raises few concerns, including:
- The BLS900 15th percentile threshold is climatologically arbitrary. The text states "there is a clear divide," but no histogram or statistical justification is shown.
- Fog detection is partially based on VK humidity ≥ 90%, even though the weather station is 3 km away and not colocalized. Fog is extremely local; this may introduce classification errors.
- The exclusion of precipitation from the fog definition removes the possibility to train the model on precipitation fog, later causing systematic false negatives.
- The qualitative validation via time-lapse imagery is not sufficiently extensive to quantify accuracy or bias.
To deal with these concerns, it is recommended that the following will be Included:
  - ROC or sensitivity analysis for percentile selection.
  - A confusion table comparing the reference against manual labeling from a subsample of time-lapse images.
  - A discussion of bias induced by using VK humidity for local fog inference.

2. **Experiments and Results.** Some interpretations mix meteorological reasoning with speculative hypotheses. For example, the drop in microwave attenuation during dense fog is attributed to droplet size changes, but no microphysical evidence is presented. Also, claims about Rayleigh theory and droplet spectra require stronger

linkage to literature[2]. It is therefore recommended to state explicitly when results are hypotheses vs validated explanations.

Two fundamentally different failure modes appear: "fog not detected due to physics" vs "fog not detected due to model design". This distinction between physics-limited detection

(e.g., microwave attenuation negligible for precipitation-fog or cold-fog due to droplet sizes) and model architecture limitations (e.g., nighttime drizzle misclassified as fog due to learned temporal priors) should be emphasized.

Please elaborate on the reduced model – how did you relate the 15 minutes min-max measurements with the instantaneous measurements in the reference data set?

Please clarify the country-wide results as presented in e.g., Fig. 14. The captions says that it presents "spatial maps comparing **fog** detection" while the reference map (b) indicates in **rain**.

3. **Discussion**

- Since both fog and humidity cause similar attenuation, the challenge of distinguishing between them in the reference dataset should be discussed, given that relative humidity higher than 90% is required

- Discussion of the potential of using other ML methods (LSTM, GRU, Temporal CNN with causal convolutions) is missing, especially since treating windows as independent loses crucial information and encourages overfitting to instantaneous patterns.

4. **Clarity/Structure**

- Figures 7–10 are information dense; consider adding small labeled boxes showing fog/non-fog intervals visually.

- The introduction reviews the fog prediction/remotesensing literature well, but ML-for-atmospheric sensing needs additional contextualization.

- Section 4.3 (Methods) is long and mixes speculation, results, and suggestions. Consider splitting into "Limitations" and "Future Work."

5. **References**

Relevant references are from 3 groups: meteorological (e.g., footnote 2), relevant machine learning, and relevant CMLs studies (e.g., footnote 1). Since the claim contribution is mostly in the use of ML for fog detection using CML, it is recommended also to mention the first[3] to use ML with CMLs measurements.
* * *
[2] For example, see Gultepe et al. (2017). *Fog Microphysics and Visibility.*

[3] Habi, Hai Victor, and Hagit Messer. "Wet-dry classification using lstm and commercial microwave links." *2018 IEEE 10th sensor array and multichannel signal processing workshop (SAM)*. IEEE, 2018.